# Risk of and duration of protection from SARS-CoV-2 reinfection assessed with real-world data

Shannon L. Reynolds[1]*, Harvey W. Kaufman[2], William A. Meyer, III[2], Chris Bush[1], Oren Cohen[3], Kathy Cronin[4], Carly Kabelac[1], Sandy Leonard[5], Steve Anderson[3], Valentina Petkov[4], Douglas Lowy[4], Norman Sharpless[4], Lynne Penberthy[4]

1 Science and Delivery, Aetion, Inc., New York, New York, United States of America, 2 Medical Affairs, Quest Diagnostics, Secaucus, New Jersey, United States of America, 3 Office of the Chief Medical Office, Labcorp Drug Development, Burlington, North Carolina, United States of America, 4 Division of Cancer Control and Population Science, National Cancer Institute, Bethesda, Maryland, United States of America, 5 Parternships and Real World Data, HealthVerity, Philadelphia, PA, United States of America

* shannon.reynolds@aetion.com

## Abstract

This retrospective observational study aimed to gain a better understanding of the protective duration of prior SARS-CoV-2 infection against reinfection. The objectives were two-fold: to assess the durability of immunity to SARS-CoV-2 reinfection among initially unvaccinated individuals with previous SARS-CoV-2 infection, and to evaluate the crude SARS-CoV-2 reinfection rate and associated risk factors. During the pandemic era time period from February 29, 2020, through April 30, 2021, 144,678,382 individuals with SARS-CoV-2 molecular diagnostic or antibody test results were studied. Rates of reinfection among index-positive individuals were compared to rates of infection among index-negative individuals. Factors associated with reinfection were evaluated using multivariable logistic regression. For both objectives, the outcome was a subsequent positive molecular diagnostic test result. Consistent with prior findings, the risk of reinfection among index-positive individuals was 87% lower than the risk of infection among index-negative individuals. The duration of protection against reinfection was stable over the median 5 months and up to 1-year follow-up interval. Factors associated with an increased reinfection risk included older age, comorbid immunologic conditions, and living in congregate care settings; healthcare workers had a decreased reinfection risk. This large US population-based study suggests that infection induced immunity is durable for variants circulating pre-Delta predominance.

## Introduction

As of October 2022, over 618 million confirmed cases of COVID-19 have been diagnosed globally [1]. Survivors have a risk of reinfection that can be associated with serious clinical outcomes [2,3]. Our previous study of a national US cohort found that seropositivity was associated with reduced risk of subsequent infection over a relatively short interval with a median of 54 days (IQR: 17 to 92 days) [4]. That study demonstrated that at >90 days after an index

**Data Availability Statement:** Data underlying the study cannot be made publicly available due to ethical concerns about patient confidentiality as well as legal concerns as the data cohort consists

of data owned by multiple third-parties. The data underlying the study are de-identified, HIPAA compliant via expert determination and licensed via HealthVerity. Data may be made available to qualified researchers on request to Info@HealthVerity.com.

**Funding:** Aetion was funded by the National Cancer Institute to conduct this study. The funders had a role in study design, interpretation of results, decision to publish and preparation of the manuscript.

**Competing interests:** Shannon L. Reynolds, Carly Kabelac, and Christopher Bush are employees of and own stock in Aetion, Inc. Harvey W. Kaufman and William A. Meyer III are employees of and own stock in Quest Diagnostics. Oren Cohen and Steve Anderson are employees of and own stock in Labcorp Drug Development. Steve Anderson has received consulting fees from Luminex. Sandy Leonard is an employee of and owns stock in HealthVerity. Douglas Lowry has received royalty-related payments from NIH. Kathy Cronin, Valentina Petkov, Norman Sharpless, and Lynne Penberthy report no conflict of interests. This does not alter our adherence to PLOS ONE policies on sharing data and materials.

SARS-CoV-2 antibody test, the ratio of positive nucleic acid amplification test (NAAT) results between those who were index SARS-CoV-2 positive versus index negative individuals was 0.1, suggesting prior infection provided ~90% protection from reinfection. Similar results were observed using real-world data (RWD) and different study designs [5–10]. Research has also demonstrated that serum SARS-CoV-2 neutralizing antibodies and T cell immunity correlate with protection against infection and reinfection [12,13].

Previous studies relying on RWD indicate prior SARS-CoV-2 infection may offer protection for at least 7 months [5–10]. One such analysis observed a rate of reinfection of 6.7 per 1,000 person-years 18 months after the initial infection [11]. Additionally, two recent systematic reviews of the efficacy of natural immunity and the duration of both natural and vaccine-induced immunity have been published [12,13]. Among the 10 studies reviewed in Kojima *et al.*, a consistently decreased risk in reinfection was observed in individuals with either natural or vaccine-induced immunity [12]. The weighted average risk reduction in repeat SARS-CoV-2 infections was 90.4% with protection observed for up to 10 months. This level of protection, and its duration, seemed nondifferential between those with a prior infection compared to vaccinated individuals. Findings from the analyses included in systematic review conducted by Pilz *et al.* were similar; immunity acquired after SARS-CoV-2 infection is highly effective and conveys protection for over 1 year, while those with "hybrid" immunity from both prior infection and SARS-CoV-2 vaccination conferred the most protection, though the duration is less understood given fewly published studies [13]. Lastly, previous research has suggested patients in certain settings (e.g., long-term care facilities, healthcare workers), the immunocompromised, or those with specific comorbid conditions may be at increased risk of reinfection than other populations, but risk factors of SARS-COV-2 reinfection are less well understood [12,13].

Large-scale RWD offer an opportunity to study patterns of infection and reinfection, available longitudinally at the individual level, making it possible to study the experiences of a seropositive population with COVID-19 in near-real time. Further advantages of real-world database studies include maximizing sample size, the ability to evaluate risk within subgroups and specific patient characteristics, and greater data capture of the patient experience over time. Cohort studies leveraging RWD can supplement prior findings of protective immunity against SARS-CoV-2 reinfection through studying a larger sample of patients with potentially longer follow-up durations. Additionally, access to patients' insurance covered medical care history means cohort studies in RWD can identify additional risk-factors for reinfection beyond those previously described, filling a critical gap in our understanding of SARS-CoV-2. The primary objective of this study was to estimate the duration of protection provided against laboratory confirmed SARS-CoV-2 reinfection among index SARS-CoV-2 positive individuals compared to the risk of infection among index SARS-CoV-2 negative individuals during the pre-vaccine COVID-19 era. The secondary objective was to estimate the rate of reinfection among those previously identified as SARS-CoV-2 positive and evaluate demographic and comorbid characteristics associated with the risk of SARS-CoV-2 reinfection.

## Materials and methods

### Data sources

The study population began with 144,678,382 individuals derived from US RWD sources curated by HealthVerity and with records of medical services obtained from February 29, 2020, through April 30, 2021. This dataset aggregates multiple unique record sources across commercial laboratory databases (including an estimated 60% of aggregate SARS-CoV-2 testing performed in the US), medical claims (both open and closed claims), pharmacy databases

(both claims and retail), hospital chargemaster (CDM) and outpatient electronic health records (EHR). Patient records, using a unique, interoperable, de-identified token, are linked across laboratory test results, medical and pharmacy claims, CDM, and any available EHR. The data included in this study are compliant with HIPAA such that no patient can be reidentified. The study was deemed exempt by the New England Institutional Review Board (#1-9757-1).

## Study design

**Study population.** The dataset included records for individuals with an index NAAT or SARS-CoV-2 antibody test from February 29, 2020 (when SARS-CoV-2 NAAT first became available for molecular diagnostic testing), through December 9, 2020 (prior to COVID-19 vaccine introduction). Individuals included based on an antibody test result could enter starting on April 15, 2020, the earliest availability of an antibody test with US Food and Drug Administration (FDA) Emergency Use Authorization (EUA).

For the primary study objective, a cohort of SARS-CoV-2-tested individuals was identified. Individuals entered based on their first SARS-CoV-2 antibody or NAAT test result (index date). Individuals were required to have ≥12 months of pre-index continuous closed medical enrollment, claims or electronic health record (EHR) activity. Individuals with discordant test results on the same day were excluded. Additionally, those who tested SARS-CoV-2 NAAT-positive within 60 days following their initial test results (applied only to the index negative group) or were lost to follow-up during those first 60 days were excluded.

For the secondary objective, a cohort of individuals with any record of SARS-CoV-2-positive test results were identified. Individuals entered based on their first SARS-CoV-2 antibody or NAAT-positive test result from February 29, 2020, through December 9, 2020. Individuals were required to have ≥12 months of pre-index continuous closed medical enrollment or activity as described above. Individuals with discordant test results on the same day were excluded.

For both objectives, the baseline period was the 12 months prior to the index test date. All available data in this period were used to identify baseline characteristics and comorbid conditions.

**Exposure, outcome and covariates.** For the primary objective, individuals entered the cohort upon their first SARS-CoV-2 NAAT or antibody test; those with a positive result were classified as the index-positive group (i.e., established SARS-CoV-2 infection) while individuals whose first NAAT or antibody test result was negative were the index-negative group (i.e., without laboratory evidence of SARS-CoV-2 infection). For the secondary objective, the cohort consisted only of individuals who had a SARS-CoV-2 positive test result.

The outcome of interest for both objectives was a SARS-CoV-2 positive NAAT result occurring at >60 days after the index test result date. The follow-up time interval began on the 61st day post index date, rather than the patient index date, to avoid misclassifying individuals who experienced prolonged viral RNA shedding in the weeks after their initial infection as having the outcome [4,14,15]. Individuals were followed until the outcome of interest or the earliest occurrence of inpatient death, outcome of interest, end of available data, end of their medical plan enrollment or end of activity.

**Sensitivity analysis.** To demonstrate the robustness of our laboratory confirmed outcome definition, we examined International Classification of Diseases (ICD)-10 diagnosis codes U07.1 or U07.2 in addition to the laboratory test results for outcome ascertainment as ICD-10 codes are commonly used in real-world research to define occurrence of COVID-19 and may affect the outcome rate [14].

**Statistical analysis.** Variables included as baseline characteristics and model covariates are defined in the supplement and were reported descriptively (sMethods1 and sMethods2 in S1 File). Continuous variables are presented as means (with standard deviation) and/or medians (with interquartile range). Categorical variables are presented as absolute and relative frequencies. Demographic covariates were assessed on the index date. Regions are defined by the US Census Bureau [16]. Presence of comorbidities was assessed during the 12-month baseline period prior to the index date, unless specified otherwise in the supplement. For the primary objective, missing data that occurred in covariates or descriptive variables were classified using a missing data indicator.

For the primary objective, propensity score matching, estimated via logistic regression, was used to adjust for potential confounding between index-positive and index-negative individuals on the index date. Individuals were matched by propensity score on a 1:1 basis using a caliper of 1.0%. Variables included in the propensity-score model are defined in the supplement (sMethods 1 in S1 File). Covariate balance post-matching was evaluated using standardized differences with a threshold of <0.10 to indicate well-balanced differences [17].

For the primary objective, we report the crude rate of SARS-CoV-2 infection per 1,000 person-years (PY) and present cumulative incidence curves. We used Cox proportional-hazard regression to estimate the hazard ratios (HR) and 95% CI of the primary outcome for index-positive versus index-negative individuals during follow-up.

For the secondary objective, we estimated the crude rate of reinfection per 1,000 PY in the overall population and by demographic subgroups. Association between individual risk factors and reinfection was explored using a Cox proportional hazard regression model. Characteristics of interest included individual demographics and comorbid conditions. Missing data elements that occurred in characteristics of interest were classified as the most frequently occurring value for categorical variables. We prespecified a 2-tailed alpha of 0.05 to establish statistical significance. Data were analyzed using the previously validated Aetion Evidence Platform version r4.27.0.20210609 and R version 3.4.2.

## Results

For the primary objective, 27,070,023 individuals met the inclusion criteria, of which 7,501 died and 4,275,540 disenrolled between index and start of follow-up; 22,786,982 individuals started follow-up 61 days post- index date. Of these, 2,023,341 (9%) were SARS-CoV-2 index-positive and 20,763,641 (91%) were SARS-CoV-2 index-negative (Fig 1).

There were similar age and sex distributions between index-positive and index-negative individuals: a mean age of 43 and 45 years with 57% and 60% female, respectively. The index-negative population exhibited slightly worse comorbid illness (average Charlson-Quan score of 0.55 versus 0.49 for the index-positive). Median follow-up time was 149 days and 162 days for index-positive and index-negative groups, respectively. Index-positive individuals had fewer NAAT tests during follow-up than index-negative individuals (mean (SD) 0.40 (2.55) vs. 0.78 (3.29) respectively). All index-positive individuals were matched to an index-negative individual in the propensity-score model, with 2,023,341 individuals in each group. Characteristics were well balanced before and after matching (Table 1).

A total of 737,742 cases of infection were observed in the unmatched population over the follow-up period; 8,869 (1.2%) occurred in the index-positive group. The crude rate of reinfection over follow-up was substantially lower in the index-positive group (9.89 per 1,000-person year) than in the index-negative group (78.39 per 1,000 PY) (Table 2). The cumulative incidence of reinfection was 0.85% (95% CI: 0.82%, 0.88%) among index-positive individuals and for infection the rate was 6.2% (95% CI: 6.1%, 6.3%) among index-negative individuals over a follow-up of 375 days from the index date (Fig 2).

Individuals in dataset: 112,317,185

SARS-CoV-2 lab test (antibody or diagnostic): 43,966,094

365 days of baseline activity or enrollment: 27,950,356

No prior SARS-CoV-2 lab test: 27,141,944

No discordant test results on index: 27,141,944

Index SARS-CoV-2 negative with no positive result
within 60 days: 27,070,023

Final cohort: 27,070,023

Individuals beginning follow-up* on day 61:
**Index negative: 20,763,641**
**Index positive: 2,023,341**

*Individuals may be censored prior to the start of follow-up due to inpatient
death (n = 7,501), disenrollment (n = 4,275,540), or the end of data (n = 0).

**Fig 1. Study population.**

The risk of reinfection was stable over the median 5 months of follow-up and up to one year among index-positive individuals (n = 1,821,183 at 4 months, n = 859,824 at 8 months and n = 147,458 at 12 months). After adjustment for baseline demographic and comorbid characteristics, the risk of reinfection in the index-positive group was 87% lower than the risk of infection in the index-negative group (HR = 0.13, 95% CI: (0.13, 0.13)). A propensity-score matched analysis provided a nearly identical estimate of the degree and duration of protection. The adjusted risk of infection over follow-up time was relatively stable (6–8 months post-index (HR = 0.12, 95% CI: (0.12, 0.13); 8–10 months post-index (HR = 0.17, 95% CI: 0.15, 0.18); 10–12 months post-index (HR = 0.15, 95% CI: 0.13, 0.18)). Stratified by index-test type (antibody only (n = 2,617,139), NAAT only (n = 19,857,392) or both (n = 312,451), the adjusted risk of reinfection was similar to the pooled estimate (HR = 0.14, 95% CI: (0.13, 0.15) for antibody only; HR = 0.13 95% CI: (0.13, 0.13) for NAAT only; HR = 0.13, 95% CI: (0.08, 0.20) for both on index). These data suggest that SARS-CoV-2 infection provides substantial protection from reinfection for at least 5 months and up to one year from recovery.

To demonstrate the robustness of laboratory-confirmed COVID defining our outcome, a sensitivity analysis incorporating ICD-10 diagnostic codes to the primary outcome was

**Table 1. Demographics and baseline comorbidities in the 365 days prior to first SARS-CoV-2 test among a SARS-CoV-2-tested cohort and among a post-propensity score match population identified between February 29th, 2020 through December 9th, 2020.**

| | | SARS-CoV-2-Tested Cohort (Unmatched) | | Post-Propensity Score Match Population | | |
| --- | --- | --- | --- | --- | --- | --- |
| | | Index Infection Status | | Index Infection Status | | |
| | | SARS-CoV-2 Negative | SARS-CoV-2 Positive | SARS-CoV-2 Negative | SARS-CoV-2 Positive | Absolute Standardized Difference |
| | Number of Patients | 20,763,641 | 2,023,341 | 2,023,341 | 2,023,341 | |
| Month/Year of Index | March 2020; n (%) | 309,638 (1.5%) | 70,871 (3.5%) | 68,691 (3.4%) | 70,871 (3.5%) | 0.011 |
| | April 2020; n (%) | 896,476 (4.3%) | 174,686 (8.6%) | 173,248 (8.6%) | 174,686 (8.6%) | |
| | May 2020; n (%) | 2,338,119 (11.3%) | 160,709 (7.9%) | 159,011 (7.9%) | 160,709 (7.9%) | |
| | June 2020; n (%) | 3,134,791 (15.1%) | 235,642 (11.6%) | 240,118 (11.9%) | 235,642 (11.6%) | |
| | July 2020; n (%) | 3,638,457 (17.5%) | 328,306 (16.2%) | 329,697 (16.3%) | 328,306 (16.2%) | |
| | August 2020; n (%) | 2,497,164 (12.0%) | 158,746 (7.8%) | 157,084 (7.8%) | 158,746 (7.8%) | |
| | September 2020; n (%) | 2,003,157 (9.6%) | 123,785 (6.1%) | 123,669 (6.1%) | 123,785 (6.1%) | |
| | October 2020; n (%) | 2,497,162 (12.0%) | 206,390 (10.2%) | 206,924 (10.2%) | 206,390 (10.2%) | |
| | November 2020; n (%) | 2,645,679 (12.7%) | 407,630 (20.1%) | 409,793 (20.3%) | 407,630 (20.1%) | |
| | December 2020; n (%) | 802,998 (3.9%) | 156,576 (7.7%) | 155,106 (7.7%) | 156,576 (7.7%) | |
| Age[a] | Mean (sd) | 45.37 (20.94) | 43.66 (20.24) | 43.42 (20.16) | 43.66 (20.24) | 0.012 |
| | Median [IQR] | 46.00 [28.00, 61.00] | 44.00 [27.00, 59.00] | 44.00 [27.00, 59.00] | 44.00 [27.00, 59.00] | |
| Age Categories[a] | 0–4; n (%) | 323,404 (1.6%) | 25,370 (1.3%) | 26,305 (1.3%) | 25,370 (1.3%) | 0.014 |
| | 5–10; n (%) | 529,691 (2.6%) | 49,455 (2.4%) | 50,372 (2.5%) | 49,455 (2.4%) | |
| | 11–15; n (%) | 566,451 (2.7%) | 70,877 (3.5%) | 70,993 (3.5%) | 70,877 (3.5%) | |
| | 16–17; n (%) | 353,930 (1.7%) | 45,845 (2.3%) | 45,862 (2.3%) | 45,845 (2.3%) | |
| | 18–29; n (%) | 3,782,123 (18.2%) | 399,067 (19.7%) | 401,688 (19.9%) | 399,067 (19.7%) | |
| | 30–39; n (%) | 3,038,679 (14.6%) | 285,178 (14.1%) | 289,483 (14.3%) | 285,178 (14.1%) | |
| | 40–49; n (%) | 2,875,685 (13.8%) | 306,387 (15.1%) | 308,039 (15.2%) | 306,387 (15.1%) | |
| | 50–64; n (%) | 5,171,959 (24.9%) | 517,540 (25.6%) | 515,418 (25.5%) | 517,540 (25.6%) | |
| | 65–74; n (%) | 2,504,778 (12.1%) | 202,023 (10.0%) | 198,547 (9.8%) | 202,023 (10.0%) | |
| | 75–84; n (%) | 1,082,811 (5.2%) | 82,636 (4.1%) | 79,130 (3.9%) | 82,636 (4.1%) | |
| | > = 85; n (%) | 532,504 (2.6%) | 38,689 (1.9%) | 37,247 (1.8%) | 38,689 (1.9%) | |
| Sex[b] | Female; n (%) | 12,491,533 (60.2%) | 1,162,459 (57.5%) | 1,164,972 (57.6%) | 1,162,459 (57.5%) | 0.003 |
| | Male; n (%) | 8,269,686 (39.8%) | 860,713 (42.5%) | 858,191 (42.4%) | 860,713 (42.5%) | |
| Insurance | Other or Unknown Insurance; n (%) | 13,351,117 (64.3%) | 1,185,231 (58.6%) | 1,190,371 (58.8%) | 1,185,231 (58.6%) | 0.025 |
| | Commercial; n (%) | 4,924,029 (23.7%) | 521,501 (25.8%) | 524,664 (25.9%) | 521,501 (25.8%) | |
| | Medicare Advantage; n (%) | 335,233 (1.6%) | 33,589 (1.7%) | 27,611 (1.4%) | 33,589 (1.7%) | |
| | Medicaid; n (%) | 2,153,262 (10.4%) | 283,020 (14.0%) | 280,695 (13.9%) | 283,020 (14.0%) | |
| Geographic Region | Northeast; n (%) | 5,765,112 (27.8%) | 511,139 (25.3%) | 490,536 (24.2%) | 511,139 (25.3%) | 0.024 |
| | Midwest; n (%) | 3,210,459 (15.5%) | 368,391 (18.2%) | 374,743 (18.5%) | 368,391 (18.2%) | |
| | South; n (%) | 7,533,647 (36.3%) | 758,303 (37.5%) | 768,518 (38.0%) | 758,303 (37.5%) | |
| | West; n (%) | 4,030,702 (19.4%) | 361,860 (17.9%) | 365,669 (18.1%) | 361,860 (17.9%) | |
| | Other/Missing; n (%) | 223,721 (1.1%) | 23,648 (1.2%) | 23,875 (1.2%) | 23,648 (1.2%) | |
| Residence or Site of Care | Congregate Care; n (%) | 18,365 (0.1%) | 1,779 (0.1%) | 1,588 (0.1%) | 1,779 (0.1%) | 0.003 |
| | SNF; n (%) | 173,704 (0.8%) | 16,849 (0.8%) | 13,507 (0.7%) | 16,849 (0.8%) | 0.019 |
| Healthcare Worker | Yes; n (%) | 27,734 (0.1%) | 3,778 (0.2%) | 3,681 (0.2%) | 3,778 (0.2%) | 0.001 |
| Index Test Type | Diagnostic Test; n (%) | 18,373,796 (88.5%) | 1,796,047 (88.8%) | 1,801,091 (89.0%) | 1,796,047 (88.8%) | 0.008 |
| | Antibody Test; n (%) | 2,699,093 (13.0%) | 230,497 (11.4%) | 225,405 (11.1%) | 230,497 (11.4%) | 0.008 |

(*Continued*)

**Table 1.** (Continued)

| | | SARS-CoV-2-Tested Cohort (Unmatched) | | Post-Propensity Score Match Population | | |
|---|---|---|---|---|---|---|
| Charlson-Quan Score, 365 Days | Mean (sd) | 0.55 (1.32) | 0.49 (1.23) | 0.45 (1.14) | 0.49 (1.23) | 0.037 |
| | Median [IQR] | 0.00 [0.00, 0.00] | 0.00 [0.00, 0.00] | 0.00 [0.00, 0.00] | 0.00 [0.00, 0.00] | |
| | Data Unavailable in Pre-index; n (%) | 2,776,487 (13.4%) | 306,165 (15.%) | 272,690 (13.5%) | 306,165 (15.%) | 0.047 |
| Additional Comorbidities | Acute and Unspecified Renal Failure; n (%) | 283,901 (1.4%) | 24,592 (1.2%) | 18,996 (0.9%) | 24,592 (1.2%) | 0.027 |
| | Asthma; n (%) | 1,170,702 (5.6%) | 102,339 (5.1%) | 96,900 (4.8%) | 102,339 (5.1%) | 0.012 |
| | Chronic Obstructive Pulmonary Disease; n (%) | 688,608 (3.3%) | 43,714 (2.2%) | 37,728 (1.9%) | 43,714 (2.2%) | 0.021 |
| | Coronary Heart Disease; n (%) | 752,595 (3.6%) | 61,019 (3.0%) | 53,579 (2.6%) | 61,019 (3.0%) | 0.022 |
| | Hypertension; n (%) | 4,036,841 (19.4%) | 388,738 (19.2%) | 373,436 (18.5%) | 388,738 (19.2%) | 0.019 |
| | Immunity Disorders; n (%) | 109,911 (0.5%) | 7,697 (0.4%) | 6,097 (0.3%) | 7,697 (0.4%) | 0.014 |
| Additional Comorbidities | Ischemic Heart Disease; n (%) | 889,768 (4.3%) | 73,185 (3.6%) | 63,154 (3.1%) | 73,185 (3.6%) | 0.027 |
| | Metabolic Syndrome; n (%) | 526,644 (2.5%) | 59,284 (2.9%) | 55,236 (2.7%) | 59,284 (2.9%) | 0.012 |
| | Pneumonia; n (%) | 399,919 (1.9%) | 49,613 (2.5%) | 40,147 (2.0%) | 49,613 (2.5%) | 0.032 |
| | Vitamin D Deficiency; n (%) | 1,440,282 (6.9%) | 144,356 (7.1%) | 135,246 (6.7%) | 144,356 (7.1%) | 0.018 |
| | Comorbidity Data Unavailable in Pre-index; n (%) | 2,206,029 (10.6%) | 243,945 (12.1%) | 216,445 (10.7%) | 243,945 (12.1%) | 0.043 |
| | Underweight; n (%) | 110,507 (0.5%) | 8,995 (0.4%) | 9,169 (0.5%) | 8,995 (0.4%) | 0.001 |
| Weight | Normal weight; n (%) | 1,498,821 (7.2%) | 118,912 (5.9%) | 117,160 (5.8%) | 118,912 (5.9%) | 0.004 |
| | Overweight; n (%) | 1,931,157 (9.3%) | 184,817 (9.1%) | 180,956 (8.9%) | 184,817 (9.1%) | 0.007 |
| | Obesity; n (%) | 2,991,072 (14.4%) | 330,400 (16.3%) | 325,109 (16.1%) | 330,400 (16.3%) | 0.007 |
| | Obesity; n (%) | 2,991,072 (14.4%) | 330,400 (16.3%) | 325,109 (16.1%) | 330,400 (16.3%) | 0.007 |

[a] Among the SARS-CoV-2-Tested Cohort, age was missing for 1,626 index negative and 274 index positive individuals. Among the post-propensity score match population, age was missing for 257 index negative and 274 index positive individuals.

[b] Among the SARS-CoV-2-Tested Cohort, sex was missing/unknown for 2,422 index negative and 169 index positive individuals. Among the post-propensity score match population, sex was missing/unknown for 178 index negative and 169 index positive individuals.

**Table 2. Rate of SARS-CoV-2 positive diagnostic tests in follow-up (April 30th, 2020 through April 30th, 2021) stratified by index SARS-CoV-2 infection status with rate ratio and hazard ratio estimates among SARS-CoV-2 positive individuals vs. SARS CoV-2 negative individuals.**

| | SARS-CoV-2-Tested Cohort (Unmatched) | | Post-Propensity Score Match Population | |
|---|---|---|---|---|
| | Index Infection Status | | Index Infection Status | |
| Parameter | SARS-CoV-2 Negative | SARS-CoV-2 Positive | SARS-CoV-2 Negative | SARS-CoV-2 Positive |
| Number of Patients | 20,763,641 | 2,023,341 | 2,023,341 | 2,023,341 |
| Number of Person-Years | 9,298,000 | 897,023 | 885,825 | 897,023 |
| Number of SARS-CoV-2 Positive Diagnostic Tests in Follow-up | 728,872 | 8,869 | 66,603 | 8,869 |
| Rate of SARS-CoV-2 Positive Diagnostic Tests in Follow-up per 1,000 Person-Years | 78.39 | 9.89 | 75.19 | 9.89 |
| Rate Ratio (vs. referent; 95% CI) | Referent | 0.13 (0.12, 0.13) | Referent | 0.13 (0.13, 0.13) |
| Adjusted Hazard Ratio[1] (vs. referent; 95% CI) | Referent | 0.13 (0.13, 0.13) | Referent | 0.13 (0.13, 0.13) |

[a] The following variables were adjusted for in the fully-adjusted model that was run on the unmatched population and included as covariates in calculating the propensity score for each individual: Month/Year of Index, Age, Gender, Insurance Category, U.S. Region, Congregate Care, SNF, Healthcare Worker, Diagnostic Test on Index, Antibody Test on Index, Charlson-Quan Score Over Prior 365 Days, Pneumonia, Asthma, Chronic Obstructive Pulmonary Disease, Immunity Disorders, Acute and Unspecified Renal Failure, Ischemic Heart Disease, Hypertension, Coronary Heart Disease, Metabolic Syndrome, Vitamin D Deficiency, Underweight, Normal Weight, Overweight, Obesity.

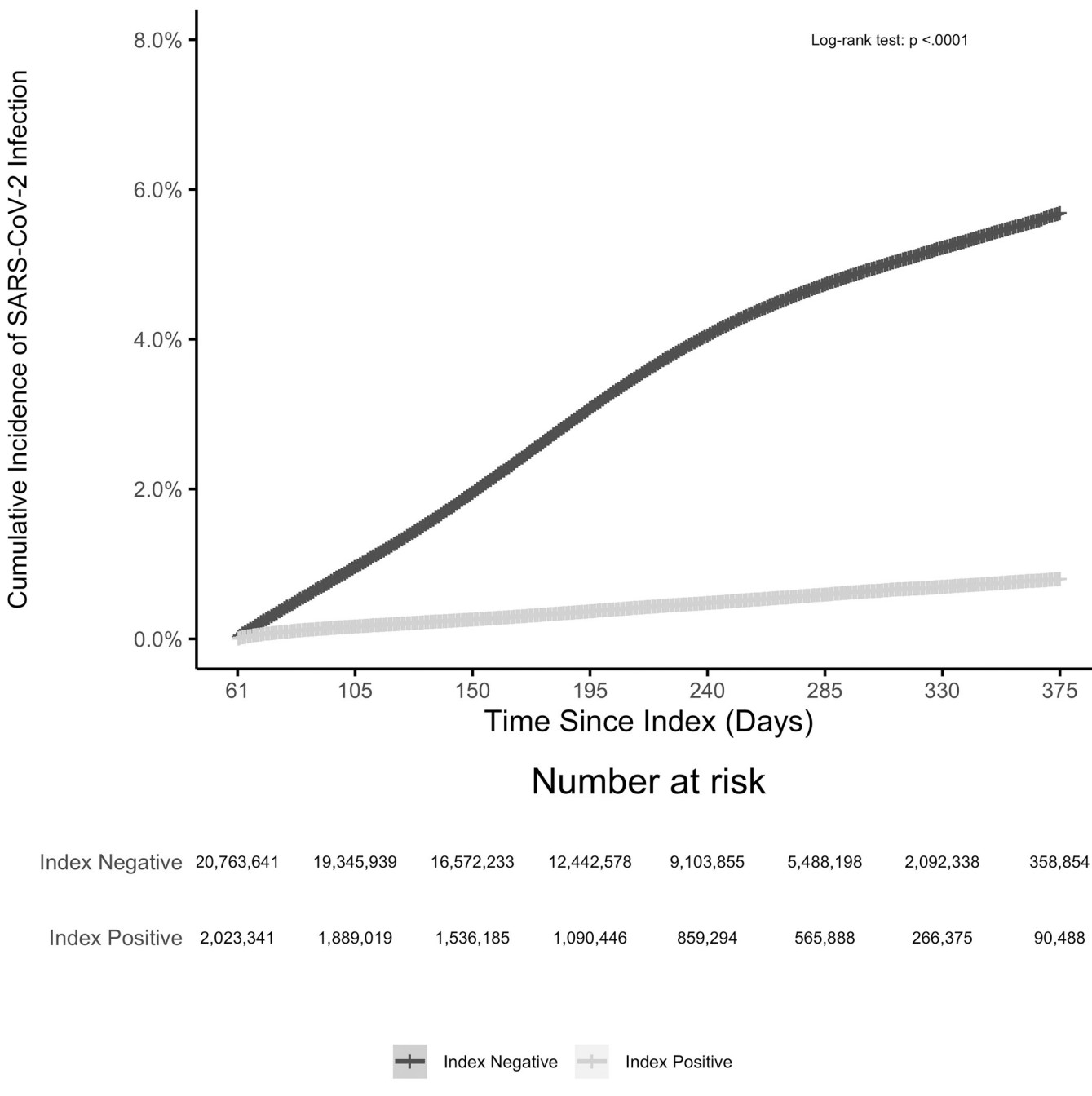

**Fig 2. Cumulative incidence of SARS-CoV-2 infection from April 30th, 2020 through April 30th, 2021 among SARS-CoV-2 index positive vs SARS-CoV-2 index negative individuals.** There were 71,921 individuals who indexed negative but had a subsequent positive test result within the following 60 days. Of the 27,070,023 index positive and negative individuals, within the first 60 days from index date, 7,501 died in an inpatient setting and 4,275,540 were disenrolled or had no additional claims or EHR activity resulting in 22,786,982 who started follow-up on day 61.

performed in the primary analysis cohort. The direction of the association was consistent with the primary analysis; however, we estimated only a 27% reduction (fully-adjusted HR = 0.73 (95% CI: 0.72, 0.73)) in risk of reinfection among index-positive individuals versus infection among index-negative individuals, compared to an 87% reduction in the main analysis.

The 60 day interval between a first positive test and assessing for the reinfection outcome was chosen to maximize follow-up data as compared to a 90 day interval, which has most often been used in prior studies [18–20]. To test the robustness of the prolonged viral shedding period definition on outcome estimates and to align to other previously published literature, we examined a 90-day exclusion period in addition to the primary definition of 60 days [18–20]. Findings (not shown) were nearly identical to the primary definition results indicating minimal bias in 60 versus 90-day viral shedding exclusion period definitions.

For the secondary study objective, 3,213,214 individuals met the criteria for the SARS-CoV-2-positive cohort, of whom 2,535,887 individuals were not censored due to death, disenrollment, or the end of data prior to beginning follow-up 61 days after their index date. The mean age was 44 years, 58.1% were female and the average Charlson-Quan score was 0.51 (Table 3). Median follow-up time was 155 days. Characteristics were similar between patients who were reinfected and those who were not.

During follow-up, the crude rate of reinfection was 11.75 (CI: 11.55, 11.96) per 1,000 PY; 12,642 cases over 1,075,563 PY. Individuals ≥85 years of age were more likely to be reinfected compared to those aged 18–29 years. This was particularly true in the Midwest and South regions. Additionally, individuals living in a skilled nursing facility (SNF) or in congregate care settings were 1.5 and 2.8 times as likely to be reinfected compared to those not residing in these settings on the index date, respectively. In the South and Midwest geographies, those in congregate care settings had a 4-times higher risk compared to those not living in congregate care settings. Patients with comorbid immunologic conditions had a 37% higher risk of being reinfected than those without. The risk of reinfection among other comorbid conditions, such as heart failure, varied by geographic region. Healthcare workers were less than half as likely to be reinfected compared to the general population (Table 4). These results suggest that immunity to SARS-CoV-2 reinfection may be less durable in elderly patients, those living in congregate setting, and in individuals with impaired immune function.

## Discussion

Among a cohort of >22 million individuals with SARS-CoV-2 laboratory test results during the pre-vaccine era, the risk of reinfection in index-positive individuals was 87% lower compared to risk of infection among index-negative individuals. This association was consistent in unadjusted, fully-adjusted, and among propensity-score matched model estimates. The observed protection against reinfection was durable for at least one year. Factors associated with an increased likelihood of reinfection included older age, comorbid immunological conditions, and living in congregate care settings, supplementing a current gap in the understanding of predictors of reinfection that have not been previously assessed utilizing large-scale RWD. Consistent with other studies, healthcare workers were half as likely to be reinfected compared to the general population, which may be associated with lifestyle factors and/or activities to reduce transmission, such as social distancing and use of mask wearing [21–23].

Our prior work suggested 90% protection in a real-world cohort of 3.2 million individuals [4]. In this current analysis, the observed magnitude of the decreased risk of infection is consistent with other published estimates in a much larger sample of patients [12,13,18,22,24]. This work adds to our prior findings showing that the real-world use of widely available diagnostics (antibody assays and NAAT) can identify individuals with prior infection and reliably predict long-term risk of reinfection.

Consistent with findings from this study, in a US study among over 325,000 patients from a health system spanning two states who were PCR tested for SARS-CoV-2 between March 2020 and September 2021, the duration of long-term protection afforded from primary SARS-CoV-

**Table 3. Demographics and baseline comorbidities in the 365 days prior to first SARS-CoV-2 positive test among a SARS-CoV-2-positive cohort identified between February 29th, 2020 through December 9th, 2020.**

| | | SARS-CoV-2-Positive Cohort | Status of Reinfection (SARS-CoV-2 Positive Diagnostic Test Over Follow-up) | |
| --- | --- | --- | --- | --- |
| | | | Reinfection | No Reinfection |
| | Number of Patients | 2,535,887 | 12,642 | 2,523,245 |
| | | | | |
| Age[a] | Mean (sd) | 43.94 (20.20) | 45.6 (20.0) | 43.9 (20.2) |
| | Median [IQR] | 44.00 [27.00, 59.00] | 45.0 [1.00, 93.0] | 44.0 [0, 93.0] |
| Age Categories[a] | <18; n (%) | 216,344 (8.5%) | 549 (4.3%) | 215,795 (8.6%) |
| | 18–29; n (%) | 513,493 (20.2%) | 2,756 (21.8%) | 510,737 (20.2%) |
| | 30–39; n (%) | 364,400 (14.4%) | 1,900 (15.0%) | 362,500 (14.4%) |
| | 40–49; n (%) | 384,973 (15.2%) | 1,991 (15.7%) | 382,982 (15.2%) |
| | 50–64; n (%) | 645,927 (25.5%) | 3,329 (26.3%) | 642,598 (25.5%) |
| | 65–74; n (%) | 250,602 (9.9%) | 1,091 (8.6%) | 249,511 (9.9%) |
| | 75–84; n (%) | 104,488 (4.1%) | 520 (4.1%) | 103,968 (4.1%) |
| | > = 85; n (%) | 55,660 (2.2%) | 506 (4.0%) | 55,154 (2.2%) |
| Sex[b] | Male; n (%) | 1,061,341 (41.9%) | 5,189 (41.0%) | 1,056,152 (41.9%) |
| | Female; n (%) | 1,474,345 (58.1%) | 7,452 (58.9%) | 1,466,893 (58.1%) |
| | Unknown; n (%) | 201 (0.0%) | 1 (0.0%) | 200 (0.0%) |
| Insurance | Other or Unknown Insurance; n (%) | 1,507,842 (59.5%) | 8,313 (65.8%) | 1,499,529 (59.4%) |
| | Commerical; n (%) | 639,128 (25.2%) | 2,473 (19.6%) | 636,655 (25.2%) |
| | Medicare Advantage; n (%) | 42,781 (1.7%) | 245 (1.9%) | 42,536 (1.7%) |
| | Medicaid; n (%) | 346,136 (13.6%) | 1,611 (12.7%) | 344,525 (13.7%) |
| Geographic Region | Northeast; n (%) | 661,667 (26.1%) | 4,659 (36.9%) | 657,008 (26.0%) |
| | Midwest; n (%) | 451,237 (17.8%) | 1,175 (9.3%) | 450,062 (17.8%) |
| | South; n (%) | 924,541 (36.5%) | 3,565 (28.2%) | 920,976 (36.5%) |
| | West; n (%) | 461,319 (18.2%) | 2,949 (23.3%) | 458,370 (18.2%) |
| | Other/Missing; n (%) | 37,123 (1.5%) | 294 (2.3%) | 36,829 (1.5%) |
| Residence or Site of Care | Congregate Care; n (%) | 3,862 (0.2%) | 33 (0.3%) | 3,829 (0.2%) |
| | SNF; n (%) | 28,580 (1.1%) | 361 (2.9%) | 28,219 (1.1%) |
| Healthcare Worker | Yes; n (%) | 6,604 (0.3%) | 11 (0.1%) | 6,593 (0.3%) |
| Index Test Type | Diagnostic Test; n (%) | 2,265,614 (89.3%) | 11,387 (90.1%) | 2,254,227 (89.3%) |
| | Antibody Test; n (%) | 322,065 (12.7%) | 1,655 (13.1%) | 320,410 (12.7%) |

*(Continued)*

**Table 3.** (Continued)

| | | SARS-CoV-2-Positive Cohort | Status of Reinfection (SARS-CoV-2 Positive Diagnostic Test Over Follow-up) | |
| --- | --- | --- | --- | --- |
| | | | Reinfection | No Reinfection |
| Hospitalization Assessed 10 Days Pre-index Through Start of Follow-up | No Hospitalization; n (%) | 2,435,625 (96.0%) | 11,991 (94.9%) | 2,423,634 (96.1%) |
| | Hospitalization With COVID-related Diagnosis/Symptoms; n (%) | 77,930 (3.1%) | 508 (4.0%) | 77,422 (3.1%) |
| | ICU and Hospitalization With COVID-related Diagnosis/Symptoms; n (%) | 22,332 (0.9%) | 143 (1.1%) | 22,189 (0.9%) |
| Charlson-Quan Score, 365 Days | Mean (sd) | 0.51 (1.27) | 0.713 (1.57) | 0.510 (1.27) |
| | Median [IQR] | 0.00 [0.00, 0.00] | 0 [0, 15.0] | 0 [0, 21.0] |
| | Data Unavailable in Pre-index; n (%) | 379,244 (15.0%) | 2,091 (16.5%) | 377,153 (14.9%) |
| Additional Comorbidities | Acute and Unspecified Renal Failure; n (%) | 34,078 (1.3%) | 302 (2.4%) | 33,776 (1.3%) |
| | Asthma; n (%) | 131,875 (5.2%) | 781 (6.2%) | 131,094 (5.2%) |
| | Chronic Obstructive Pulmonary Disease; n (%) | 203,451 (8.0%) | 1,262 (10.0%) | 202,189 (8.0%) |
| | Coronary Heart Disease; n (%) | 80,403 (3.2%) | 542 (4.3%) | 79,861 (3.2%) |
| | Hypertension; n (%) | 494,021 (19.5%) | 2,819 (22.3%) | 491,202 (19.5%) |
| | Immunity Disorders; n (%) | 10,015 (0.4%) | 82 (0.6%) | 9,933 (0.4%) |
| | Ischemic Heart Disease; n (%) | 96,057 (3.8%) | 646 (5.1%) | 95,411 (3.8%) |
| | Metabolic Syndrome; n (%) | 75,362 (3.0%) | 475 (3.8%) | 74,887 (3.0%) |
| | Pneumonia; n (%) | 66,874 (2.6%) | 466 (3.7%) | 66,408 (2.6%) |
| | Vitamin D Deficiency; n (%) | 184,215 (7.3%) | 1,034 (8.2%) | 183,181 (7.3%) |
| | Comorbidity Data Unavailable in Pre-index; n (%) | 303,847 (12.0%) | 1,626 (12.9%) | 302,221 (12.0%) |
| Weight | Underweight; n (%) | 10,672 (0.4%) | 39 (0.3%) | 10,633 (0.4%) |
| | Normal weight; n (%) | 150,979 (6.0%) | 798 (6.3%) | 150,181 (6.0%) |
| | Overweight; n (%) | 231,568 (9.1%) | 1,297 (10.3%) | 230,271 (9.1%) |
| | Obesity; n (%) | 413,571 (16.3%) | 2,350 (18.6%) | 411,221 (16.3%) |

[a] Among the SARS-CoV-2-Positive Cohort, age is missing for 278 individuals. Individuals with missing age were reassigned to the most frequently occurring group—the 50–64 year old category.

[b] Among the SARS-CoV-2-Positive Cohort, sex is missing for 16 individuals. Individuals with missing sex were reassigned to the most frequently occurring group—female.

2 infection was up to 13 months [25]. A population-based study of over 500,000 individuals in Denmark evaluating infection measured by PCR testing during the second COVID-19 surge among patients who were tested in the first COVID-19 surge, suggest duration lasting at least 7 months, as no waning immunity was observed when comparing results at 3–6 months vs. ≥7 months [24]. Further, a meta-analysis conducted in 2021 suggested that immunity from primary SARS-COV-2 infection likely persists through one year [26]. This present study now extends those observations by examining a much larger, real-world population followed for over a year and corroborates findings from two systematic literature reviews.

The stable level of protection from reinfection observed in this study through the first year after SARS-CoV-2 infection differs from the duration of protection after two doses of a SARS-CoV-2 mRNA vaccine, which has been reported to decrease after a few months [27]. This

**Table 4. Measured association of demographic and clinical factors with re-infection (SARS-CoV-2 positive diagnostic test during follow-up (April 30th, 2020 through April 30th, 2021)) among individuals with history of SARS-CoV-2 infection identified between February 29th, 2020 through December 9th, 2020.**

| | | SARS-CoV-2-Positive Cohort | Northeast | Midwest | South | West |
|---|---|---|---|---|---|---|
| | | Multivariate Estimates | Multivariate Estimates | Multivariate Estimates | Multivariate Estimates | Multivariate Estimates |
| | | Hazard Ratio (95% CI) | Hazard Ratio (95% CI) | Hazard Ratio (95% CI) | Hazard Ratio (95% CI) | Hazard Ratio (95% CI) |
| Categorical Age[a] (vs. 18–29) | Aged <18 | 0.50 (0.45, 0.54) | 0.44 (0.38, 0.53) | 0.68 (0.51, 0.92) | 0.49 (0.42, 0.58) | 0.53 (0.45, 0.64) |
| | Aged 30–39 | 0.94 (0.89, 1.00) | 0.89 (0.80, 0.98) | 1.00 (0.82, 1.21) | 0.89 (0.79, 1.00) | 0.99 (0.88, 1.11) |
| | Aged 40–49 | 0.91 (0.86, 0.96) | 0.89 (0.81, 0.98) | 0.91 (0.75, 1.11) | 0.90 (0.81, 1.01) | 0.93 (0.83, 1.05) |
| | Aged 50–64 | 0.88 (0.83, 0.93) | 0.82 (0.75, 0.90) | 0.99 (0.84, 1.18) | 0.88 (0.79, 0.97) | 0.91 (0.82, 1.02) |
| | Aged 65–74 | 0.75 (0.69, 0.80) | 0.63 (0.56, 0.72) | 0.90 (0.71, 1.15) | 0.84 (0.74, 0.96) | 0.72 (0.61, 0.85) |
| | Aged 75–84 | 0.82 (0.74, 0.91) | 0.73 (0.62, 0.87) | 1.07 (0.79, 1.46) | 0.84 (0.70, 1.01) | 0.88 (0.71, 1.08) |
| | Aged 85+ | 1.29 (1.16, 1.43) | 1.04 (0.87, 1.24) | 1.77 (1.29, 2.44) | 1.59 (1.32, 1.93) | 1.12 (0.87, 1.45) |
| Sex[b] (vs. Female) | Male | 1.01 (0.97, 1.04) | 0.98 (0.92, 1.04) | 0.90 (0.80, 1.02) | 1.01 (0.94, 1.08) | 0.99 (0.92, 1.07) |
| | Unknown | 1.86 (0.26, 13.20) | 0.00 (0.00, ∞) | 0.00 (0.00, 0.00) | 7.24 (1.02, 51.4) | 0.00 (0.00, ∞) |
| Residence or Site of Care | Congregate Care | 2.80 (1.98, 3.97) | 2.18 (1.28, 3.71) | 2.00 (0.87, 4.62) | 4.11 (1.83, 9.23) | 4.74 (1.75, 12.83) |
| | SNF | 1.53 (1.35, 1.72) | 1.69 (1.39, 2.03) | 1.56 (1.08, 2.25) | 1.61 (1.28, 2.04) | 1.20 (0.90, 1.58) |
| Healthcare Worker (vs. No) | Yes | 0.48 (0.27, 0.88) | 0.51 (0.19, 1.37) | 0.91 (0.33, 2.51) | 0.21 (0.03, 1.53) | 0.27 (0.04, 1.94) |
| Hospitalization Assessed 10 Days Pre-index Through Start of Follow-up (vs. No Hospitalization) | Hospitalization With COVID-19-Related Diagnosis/Symptoms | 1.00 (0.91, 1.09) | 0.99 (0.84, 1.18) | 1.03 (0.77, 1.39) | 0.90 (0.76, 1.07) | 1.17 (0.99, 1.39) |
| | ICU and Hospitalization With COVID-19-Related Diagnosis/Symptoms | 0.91 (0.77, 1.08) | 0.82 (0.57, 1.16) | 0.58 (0.27, 1.23) | 0.76 (0.54, 1.05) | 1.06 (0.83, 1.37) |
| Comorbid Conditions | Charlson Deyo Score | 1.04 (1.00, 1.07) | 1.01 (0.96, 1.06) | 1.00 (0.87, 1.15) | 1.08 (1.02, 1.14) | 1.01 (0.94, 1.09) |
| | Acute and Unspecified Renal Failure | 1.18 (1.03, 1.35) | 1.33 (1.06, 1.65) | 0.68 (0.42, 1.10) | 1.06 (0.83, 1.36) | 1.38 (1.07, 1.78) |
| | Asthma | 1.03 (0.91, 1.15) | 0.99 (0.82, 1.21) | 0.86 (0.61, 1.23) | 0.95 (0.77, 1.17) | 1.21 (0.94, 1.57) |
| | Chronic Obstructive Pulmonary Disease | 1.02 (0.92, 1.13) | 1.05 (0.88, 1.25) | 1.27 (0.93, 1.74) | 1.09 (0.92, 1.31) | 0.87 (0.69, 1.10) |
| | Congestive Heart Failure | 1.11 (0.99, 1.24) | 1.09 (0.90, 1.31) | 1.50 (1.05, 2.14) | 1.05 (0.86, 1.29) | 1.11 (0.87, 1.43) |
| | Coronary Heart Disease | 0.97 (0.79, 1.20) | 0.90 (0.65, 1.25) | 0.64 (0.33, 1.24) | 1.00 (0.69, 1.44) | 1.50 (0.87, 2.57) |
| Comorbid Conditions | Diabetes Without Complications | 1.03 (0.96, 1.11) | 1.08 (0.96, 1.21) | 1.09 (0.84, 1.42) | 0.99 (0.87, 1.13) | 0.95 (0.81, 1.11) |
| | Diabetes With Complications | 0.99 (0.89, 1.10) | 0.87 (0.73, 1.04) | 1.10 (0.76, 1.60) | 0.91 (0.75, 1.11) | 1.24 (1.00, 1.54) |
| | History of Malignancy | 1.06 (0.94, 1.21) | 1.04 (0.85, 1.28) | 0.71 (0.42, 1.20) | 1.06 (0.84, 1.34) | 1.37 (1.05, 1.79) |
| | History of Metastatic Solid Tumor | 0.98 (0.70, 1.36) | 1.41 (0.85, 2.32) | 1.02 (0.27, 3.92) | 0.70 (0.37, 1.33) | 0.83 (0.41, 1.66) |
| | Hypertension | 0.88 (0.83, 0.93) | 0.86 (0.79, 0.94) | 0.92 (0.77, 1.10) | 0.95 (0.86, 1.05) | 0.96 (0.85, 1.08) |
| | Immunity Disorders | 1.35 (1.08, 1.68) | 1.32 (0.93, 1.88) | 1.27 (0.60, 2.69) | 1.43 (0.96, 2.12) | 1.40 (0.87, 2.24) |
| | Ischemic Heart Disease | 1.02 (0.84, 1.24) | 1.03 (0.75, 1.39) | 1.47 (0.80, 2.71) | 1.08 (0.77, 1.53) | 0.74 (0.44, 1.25) |
| | Myocardial Infarction | 0.99 (0.84, 1.18) | 1.02 (0.76, 1.38) | 1.10 (0.64, 1.90) | 0.90 (0.65, 1.24) | 0.93 (0.66, 1.33) |
| | Obesity | 1.05 (1.00, 1.10) | 1.05 (0.97, 1.14) | 0.96 (0.82, 1.13) | 0.99 (0.90, 1.09) | 1.15 (1.05, 1.26) |
| | Peripheral Vascular Disease | 1.11 (1.00, 1.22) | 1.19 (1.02, 1.39) | 1.02 (0.71, 1.48) | 1.03 (0.86, 1.24) | 1.09 (0.87, 1.37) |
| | Renal Disease | 0.96 (0.85, 1.09) | 1.03 (0.84, 1.28) | 1.12 (0.72, 1.73) | 0.86 (0.70, 1.07) | 0.96 (0.74, 1.24) |
| | Rheumatic Disease | 1.07 (0.92, 1.23) | 0.97 (0.76, 1.23) | 1.18 (0.73, 1.92) | 1.23 (0.96, 1.57) | 0.98 (0.71, 1.33) |
| | Stroke | 1.03 (0.92, 1.15) | 1.09 (0.92, 1.29) | 0.93 (0.62, 1.40) | 0.97 (0.79, 1.18) | 1.07 (0.83, 1.38) |

*(Continued)*

**Table 4.** (Continued)

| | | SARS-CoV-2-Positive Cohort | Northeast | Midwest | South | West |
|---|---|---|---|---|---|---|
| | | **Multivariate Estimates** | **Multivariate Estimates** | **Multivariate Estimates** | **Multivariate Estimates** | **Multivariate Estimates** |
| | | **Hazard Ratio (95% CI)** | **Hazard Ratio (95% CI)** | **Hazard Ratio (95% CI)** | **Hazard Ratio (95% CI)** | **Hazard Ratio (95% CI)** |
| Month of Index (vs. July 2020) | March 2020 | 1.15 (1.05, 1.26) | 0.78 (0.68, 0.89) | 0.62 (0.39, 0.99) | 1.46 (1.18, 1.79) | 0.95 (0.71, 1.27) |
| | April 2020 | 1.29 (1.21, 1.37) | 0.82 (0.73, 0.92) | 1.11 (0.88, 1.41) | 1.94 (1.70, 2.22) | 1.21 (1.02, 1.44) |
| | May 2020 | 1.12 (1.05, 1.19) | 0.73 (0.65, 0.82) | 0.79 (0.63, 1.01) | 1.62 (1.42, 1.85) | 1.04 (0.88, 1.21) |
| | June 2020 | 1.02 (0.96, 1.08) | 0.77 (0.67, 0.89) | 0.95 (0.75, 1.20) | 1.06 (0.95, 1.17) | 1.08 (0.97, 1.20) |
| | August 2020 | 1.15 (1.07, 1.24) | 1.08 (0.92, 1.27) | 0.86 (0.69, 1.07) | 1.15 (1.03, 1.29) | 1.24 (1.09, 1.42) |
| | September 2020 | 1.18 (1.09, 1.28) | 1.16 (0.98, 1.37) | 0.76 (0.60, 0.96) | 1.20 (1.05, 1.38) | 1.26 (1.08, 1.47) |
| | October 2020 | 0.94 (0.87, 1.01) | 1.13 (0.97, 1.32) | 0.51 (0.41, 0.64) | 0.97 (0.84, 1.11) | 0.91 (0.78, 1.07) |
| | November 2020 | 0.76 (0.70, 0.81) | 0.97 (0.85, 1.12) | 0.31 (0.25, 0.38) | 0.81 (0.71, 0.92) | 0.72 (0.64, 0.83) |
| | December 2020 | 0.77 (0.69, 0.85) | 0.93 (0.78, 1.12) | 0.31 (0.22, 0.43) | 0.71 (0.57, 0.88) | 0.65 (0.53, 0.78) |

[a] Among the SARS-CoV-2-Positive Cohort, age is missing for 278 individuals. Individuals with missing age were reassigned to the most frequently occurring group—the 50–64 year old category.

[b] Among the SARS-CoV-2-Positive Cohort, sex is missing for 16 individuals. Individuals with missing sex were reassigned to the most frequently occurring group—female.

[c] Regions were defined as the following: Northeast:{Connecticut, Maine, Massachusetts, New Hampshire, Rhode Island, Vermont, New Jersey, New York, Pennsylvania}, Midwest: {Indiana, Illinois, Michigan, Ohio, Wisconsin, Iowa, Kansas, Minnesota, Missouri, Nebraska, North Dakota, South Dakota}, South: {Delaware, District of Columbia, Florida, Maryland, North Carolina, South Carolina, Virginia, West Virginia, Alabama, Kentucky, Mississippi, Tennessee, Arkansas, Lousiana, Oklahoma, Texas, Georgia,}, West: {Arizona, Colorado, Idaho, New Mexico, Montana, Utah, Nevada, Wyoming, Alaska, California, Hawaii, Oregon, Washington}.

difference may not be surprising, given what is known about duration of protection from other subunit vaccines compared with duration of protection following viral infection, and the much more rapid reported decrease in antibody titer following two doses of a SARS-CoV-2 mRNA vaccine compared with the rate of decrease after viral infection [28,29].

Of importance to future studies utilizing real-world evidence and diagnostic codes where more reliable laboratory data aren't readily available as in our study, when ICD-10 diagnostic codes were added to the primary outcome definition, we found a notably lower risk reduction (27%) for reinfection vs. infection. This suggests the lack of specificity in COVID-19 ICD-10 codes. Other studies have found variable positive predictive value of COVID ICD-10 codes based on care setting (i.e., inpatient or outpatient) [30–33].

The present study identified a large national population using data from medical and pharmacy claims, retail pharmacy data, and electronic medical records. This large size and broad representation across many evaluated attributes allowed better characterization of subgroups. These data are not specifically intended for research purposes; thus, the completeness of medical information is unknown. Additionally, certain risk factors for infection, such as frequency of exposure to SARS-CoV-2 are not captured in the RWD. We were, therefore, not able to assess social and behavioral factors that likely influence risk of reinfection, which may be why we observe the same effect estimates in the crude and adjusted models.

These data are mainly drawn from a medically insured individuals, except for those laboratory or retail pharmacy data coming directly from available clinical laboratory and retail pharmacy sources. As such, these data may not be representative of the medically uninsured individuals. During follow-up, index negative individuals had slightly more follow-up time

and NAAT tests than index-positive, which may overestimate the true protective effect observed. Further, false-negative NAAT test results among the index-negative may underestimate the true protective effect observed [34]. Study data are not inclusive of time periods of more recent variant circulation (i.e. Delta, Omicron); durability of protection may vary for these and future SARS-CoV-2 variants. Additionally, individuals' vaccination status over follow-up was not adjusted for in this analysis, however a sensitivity analysis (results not shown) limiting the study period to the pre-COVID-19 vaccine era demonstrated directionally similar estimates to the main analysis.

The primary outcome definition inherently required individuals to be observable, and without evidence of positive test results if in the index-negative group, until 61 days following their index date, thereby introducing immortal time bias (i.e. bias in the estimator due to exclusion of time intervals, in this case 60 days post-index). However, this decision was warranted to ensure residual viral shedding was not captured during a primary infection, creating a more specific outcome definition [9,14,15].

In summary, this large US population-based study demonstrates that SARS-CoV-2 reinfection is uncommon among individuals with laboratory evidence of a previous infection during the pre-vaccine era. Protection from SARS-CoV-2 reinfection is stable for up to one year. Reinfection risk was primarily associated with age 85+ years, comorbid immunologic conditions and living in congregate care settings; healthcare workers demonstrated a decreased reinfection risk. These findings suggest that infection induced immunity is durable for variants circulating prior to Delta.

## Supporting information

**S1 File.**
(DOCX)

## Acknowledgments

The authors would like to acknowledge Wendy Turenne for her support on this engagement and Reyna Klesh for her data expertise support.

## Author Contributions

**Conceptualization:** Shannon L. Reynolds, Harvey W. Kaufman, William A. Meyer, III, Chris Bush, Oren Cohen, Kathy Cronin, Sandy Leonard, Steve Anderson, Valentina Petkov, Douglas Lowy, Norman Sharpless, Lynne Penberthy.

**Data curation:** Shannon L. Reynolds, Harvey W. Kaufman, William A. Meyer, III, Oren Cohen, Sandy Leonard, Steve Anderson.

**Formal analysis:** Shannon L. Reynolds, Harvey W. Kaufman, William A. Meyer, III, Chris Bush, Oren Cohen, Kathy Cronin, Carly Kabelac, Steve Anderson, Valentina Petkov, Lynne Penberthy.

**Funding acquisition:** Shannon L. Reynolds.

**Investigation:** Shannon L. Reynolds, Chris Bush, Carly Kabelac.

**Methodology:** Shannon L. Reynolds, Harvey W. Kaufman, William A. Meyer, III, Chris Bush, Oren Cohen, Kathy Cronin, Carly Kabelac, Steve Anderson, Valentina Petkov, Lynne Penberthy.

**Project administration:** Shannon L. Reynolds.

**Software:** Chris Bush, Carly Kabelac.

**Supervision:** Shannon L. Reynolds, Sandy Leonard, Douglas Lowy, Norman Sharpless, Lynne Penberthy.

**Validation:** Harvey W. Kaufman, Chris Bush, Carly Kabelac.

**Visualization:** Shannon L. Reynolds, Harvey W. Kaufman, William A. Meyer, III, Chris Bush, Oren Cohen, Kathy Cronin, Carly Kabelac, Steve Anderson, Valentina Petkov, Lynne Penberthy.

**Writing – original draft:** Shannon L. Reynolds, Harvey W. Kaufman, William A. Meyer, III, Chris Bush, Oren Cohen, Kathy Cronin, Carly Kabelac, Steve Anderson, Valentina Petkov, Lynne Penberthy.

**Writing – review & editing:** Shannon L. Reynolds, Harvey W. Kaufman, William A. Meyer, III, Chris Bush, Oren Cohen, Kathy Cronin, Carly Kabelac, Sandy Leonard, Steve Anderson, Valentina Petkov, Douglas Lowy, Norman Sharpless, Lynne Penberthy.

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
