## [Decision Letter · Decision Letter 0]

3 Nov 2022

PONE-D-22-28420Risk of and duration of protection from SARS-CoV-2 reinfection assessed with real-world dataPLOS ONE

Dear Dr. Reynolds,

Thank you for submitting your manuscript to PLOS ONE. After careful consideration, we feel that it has merit but does not fully meet PLOS ONE’s publication criteria as it currently stands. Therefore, we invite you to submit a revised version of the manuscript that addresses the points raised during the review process.

We look forward to receiving your revised manuscript.

Kind regards,

Cecilia Acuti Martellucci, M.D.

Academic Editor

PLOS ONE

Journal Requirements:

"Aetion was funded by the National Cancer Institute to conduct this study."

"Shannon L. Reynolds, Carly Kabelac, and Christopher Bush are employees of and own stock in Aetion, Inc. Harvey W. Kaufman and William A. Meyer III are employees of and own stock in Quest Diagnostics. Oren Cohen and Steve Anderson are employees of and own stock in Labcorp Drug Development. Steve Anderson has received consulting fees from Luminex. Sandy Leonard is an employee of and owns stock in HealthVerity. Douglas Lowry has received royalty-related payments from NIH. Kathy Cronin, Valentina Petkov, Norman Sharpless, and Lynne Penberthy report no conflict of interests."

Additional Editor Comments:

I believe the manuscript can be considered favourably for publication, provided that the suggestions by Reviewer 2 are addressed.

Secondly, I suggest to comment this work on the same subject, which also had a long follow-up: https://doi.org/10.3389/fpubh.2022.884121. Finally, it is advisable to provide information about the ethics clearance or its potential waiving.

Reviewers' comments:

Reviewer's Responses to Questions

**Comments to the Author**

1. Is the manuscript technically sound, and do the data support the conclusions?

Reviewer #1: Yes

Reviewer #2: Yes

2. Has the statistical analysis been performed appropriately and rigorously? 

Reviewer #1: Yes

Reviewer #2: Yes

3. Have the authors made all data underlying the findings in their manuscript fully available?

Reviewer #1: Yes

Reviewer #2: No

4. Is the manuscript presented in an intelligible fashion and written in standard English?

Reviewer #1: Yes

Reviewer #2: Yes

5. Review Comments to the Author

Reviewer #1: Thanks very much for this very interesting and informative article. I feel that overall the paper was well written and without significant flaws. The study objectives and the methods section are clearly defined, the article is easily readable, and the topic is relevant to the readership. Limitations are adequately addressed. Conclusions are appropriate for the scope of the study.

Reviewer #2: This is an interesting retrospective study on SARS-CoV-2 reinfection risk in the general population. I have only a few minor comments.

The authors should provide some data on potential vaccination against SARS-CoV-2 that may have already been the case in the beginning of 2021. If they have no data on this, they should acknowledge this as a limitation.

Do the authors have specific data on the severity of the infections (e.g. severe COVID-19, ICU admissions etc.) and whether this differed between first infections and re-infections?

The authors used 60 days and not 90days as the minimum interval between a first positive test and a re-infection. As 90 days is the CDC recommendation to diagnose re-infections the authors should explain why they used their approach.

I would suggest including a figure showing a flow chart of the participants.

The authors should describe in more detail why their study was needed and what their findings add to the existing literature.

Page 9: “HR =0.13, 95% CI: 0.13-0.13”; is the CI for the HR correct, please check.

The authors should reference the first study on this issue from Austria (Eur J Clin Invest. 2021 Apr;51(4):e13520. doi: 10.1111/eci.13520. Epub 2021 Feb 21).

6. PLOS authors have the option to publish the peer review history of their article (what does this mean?). If published, this will include your full peer review and any attached files.

Reviewer #1: **Yes: **Mumoli Nicola, Department of Internal Medicine, Magenta (MI), Italy.

Reviewer #2: No

---

## [Author Response · Author response to Decision Letter 0]

16 Dec 2022

Journal Requirements:

We have addressed the journal requirements. We have ensured the manuscript meets PLOS ONE's style requirements, including those for file naming. Additionally, we have included an updated Role of Funder statement, Competing Interests statement and Data Availability statement in our cover letter.

Academic editor comments: 

I believe the manuscript can be considered favourably for publication, provided that the suggestions by Reviewer 2 are addressed.

Secondly, I suggest to comment this work on the same subject, which also had a long follow-up: https://doi.org/10.3389/fpubh.2022.884121. Finally, it is advisable to provide information about the ethics clearance or its potential waiving.

Response: Thank you for the suggestion. We have included a reference to this study in the background section. Additionally, we have provided more information about the ethical concerns of data sharing in the Data Availability statement. 

Reviewer #1 comments: 

Thanks very much for this very interesting and informative article. I feel that overall the paper was well written and without significant flaws. The study objectives and the methods section are clearly defined, the article is easily readable, and the topic is relevant to the readership. Limitations are adequately addressed. Conclusions are appropriate for the scope of the study.

Response: Thank you for taking the time to review our manuscript. We appreciate your thoughtful feedback. 

Reviewer #2 comments: 

This is an interesting retrospective study on SARS-CoV-2 reinfection risk in the general population. I have only a few minor comments. The authors should provide some data on potential vaccination against SARS-CoV-2 that may have already been the case in the beginning of 2021. If they have no data on this, they should acknowledge this as a limitation.

Response: The dataset included records for individuals with an index NAAT or SARS-CoV-2 antibody test from February 29, 2020 through December 9, 2020, prior to vaccine availability. However, individuals could potentially be vaccinated after index. In order to understand how the conclusions from the main analysis for the primary objective could change if limited to a pre-vaccine era, a sensitivity analysis was conducted that restricted the cohort identification period to end on June 10, 2020 with follow-up ending on December 9, 2020. The direction of the association between index test result and risk of SARS-CoV-2 infection was consistent with the primary analysis. We estimated a fully adjusted hazard ratio (95% CI) of 0.25 (0.23, 0.26), indicating a 75% lower risk of infection for individuals whose index test was positive compared to those whose index test was negative, compared to an 87% lower risk in the primary analysis. We have added additional language to the limitations in the discussion section noting that individuals vaccination status’ over follow-up is not adjusted for in the main analysis and that a sensitivity analysis limiting the time period to the pre-vaccine era showed directionally similar results.

Do the authors have specific data on the severity of the infections (e.g. severe COVID-19, ICU admissions etc.) and whether this differed between first infections and re-infections?

Response: In Table 3, we reported the severity of the initial infection among a SARS-CoV-2-positive cohort. Individuals were classified into one of the following categories: no hospitalization, hospitalization with COVID-related diagnosis/symptoms, or ICU and hospitalization with COVID-related diagnosis/symptoms. We examined the severity of the initial infection among those who had a subsequent reinfection and those who did not and found that the severity of the initial infection was similar regardless of reinfection status. Among those who were reinfected, we did not examine the severity of the reinfection. 

The authors used 60 days and not 90 days as the minimum interval between a first positive test and a re-infection. As 90 days is the CDC recommendation to diagnose re-infections the authors should explain why they used their approach.

Response: To test the robustness of the prolonged viral shedding period definition on outcome estimates and to align to other previously published literature, we examined a 90-day exclusion period in addition to the primary definition of 60 days. Findings (not shown) were nearly identical to the primary definition results indicating minimal bias in 60 versus 90-day viral shedding exclusion period definitions. The 60 day interval between a first positive test and assessing for the reinfection outcome was chosen to maximize follow-up data as compared to a 90 day interval without significant impact to the effect estimate as observed by the sensitivity analysis performed. We have included the results for this sensitivity analysis as well as language explaining our choice to use a 60-day interval on page 11. 

I would suggest including a figure showing a flow chart of the participants.

Response: Thank you for the suggestion. We have included a participant flow-chart in the resubmission.

The authors should describe in more detail why their study was needed and what their findings add to the existing literature.

Response: We have addressed this in the background section by citing the references provided by the reviewers and including more language for the knowledge gap that this research addresses. Additional language has also been provided in the discussion section to emphasize the contributions of this analysis. 

Page 9: “HR =0.13, 95% CI: 0.13-0.13”; is the CI for the HR correct, please check.

Response: We confirmed that this CI is correct. The point estimate and upper and lower CI limits appear as 0.13 due to rounding to the hundredths place. If the decimal is extended out to the thousandths place, the estimate is 0.132 (0.129, 0.135). 

The authors should reference the first study on this issue from Austria (Eur J Clin Invest. 2021 Apr;51(4):e13520. doi: 10.1111/eci.13520. Epub 2021 Feb 21

Response: Thank you for the suggestion! We have included a reference to this study in the background section.

---

## [Decision Letter · Decision Letter 1]

4 Jan 2023

Risk of and duration of protection from SARS-CoV-2 reinfection assessed with real-world data

PONE-D-22-28420R1

Dear Dr. Reynolds,

We’re pleased to inform you that your manuscript has been judged scientifically suitable for publication and will be formally accepted for publication once it meets all outstanding technical requirements.

Kind regards,

Cecilia Acuti Martellucci, M.D.

Academic Editor

PLOS ONE

Additional Editor Comments (optional):

All reviewers comments were addressed and the manuscript is now fit for publication.

Reviewers' comments:

Reviewer's Responses to Questions

**Comments to the Author**

1. If the authors have adequately addressed your comments raised in a previous round of review and you feel that this manuscript is now acceptable for publication, you may indicate that here to bypass the “Comments to the Author” section, enter your conflict of interest statement in the “Confidential to Editor” section, and submit your "Accept" recommendation.

Reviewer #1: All comments have been addressed

Reviewer #2: All comments have been addressed

2. Is the manuscript technically sound, and do the data support the conclusions?

Reviewer #1: Yes

Reviewer #2: Yes

3. Has the statistical analysis been performed appropriately and rigorously? 

Reviewer #1: Yes

Reviewer #2: Yes

4. Have the authors made all data underlying the findings in their manuscript fully available?

Reviewer #1: Yes

Reviewer #2: No

5. Is the manuscript presented in an intelligible fashion and written in standard English?

Reviewer #1: Yes

Reviewer #2: Yes

6. Review Comments to the Author

Reviewer #1: After revision the manuscript is worthy to be published. I feel that overall the paper was well written and without significant flaws. The study objectives and the methods section are clearly defined, the article is easily readable, and the topic is relevant to the readership. Limitations are adequately addressed. Conclusions are appropriate for the scope of the study. The paper is formally correct and it is clear its clinical relevance, and what this article should add to the body of knowledge on this topic.

Reviewer #2: The authors have addressed all my comments. I have no further criticism and congratulate the authors for their great work!

7. PLOS authors have the option to publish the peer review history of their article (what does this mean?). If published, this will include your full peer review and any attached files.

Reviewer #1: **Yes: **Mumoli Nicola

Reviewer #2: No

---

## [Editor Report · Acceptance letter]

2 Feb 2023

PONE-D-22-28420R1 

Risk of and duration of protection from SARS-CoV-2 reinfection assessed with real-world data 

Dear Dr. Reynolds:

I'm pleased to inform you that your manuscript has been deemed suitable for publication in PLOS ONE. Congratulations! Your manuscript is now with our production department. 

Kind regards, 

on behalf of

Dr. Cecilia Acuti Martellucci 

Academic Editor

PLOS ONE